Chlorophyll enhances oxidative stress tolerance in Caenorhabditis elegans and extends its lifespan

Wang Erjia
Wink Michael wink@uni-heidelberg.de
Heidelberg University, Institute of Pharmacy and Molecular Biotechnology, Department of Biology , Heidelberg , Germany
Allison David
Electronic publication date: 2016 Apr 7
Publication date: 2016
Volume: 4
Electronic Location ID: e1879
Received 2015 Sep 30; Accepted 2016 Mar 11
Copyright: ©2016 Wang and Wink
Copyright year: 2016
Copyright holder: Wang and Wink
License: This is an open access article distributed under the terms of the Creative Commons Attribution License, which permits unrestricted use, distribution, reproduction and adaptation in any medium and for any purpose provided that it is properly attributed. For attribution, the original author(s), title, publication source (PeerJ) and either DOI or URL of the article must be cited.
License URL: https://creativecommons.org/licenses/by/4.0/

Keywords: Chlorophyll, Spinach, Lifespan, DAF-16, C. elegans, Anti-aging, DPPH, ABTS, Antioxidant

Funding: The authors received no funding for this work.

==============================
Green vegetables are thought to be responsible for several beneficial properties such as antioxidant, anti-mutagenic, and detoxification activities. It is not known whether these effects are due to chlorophyll which exists in large amounts in many foods or result from other secondary metabolites. In this study, we used the model system Caenorhabditis elegans to investigate the anti-oxidative and anti-aging effects of chlorophyll in vivo. We found that chlorophyll significantly improves resistance to oxidative stress. It also enhances the lifespan of C. elegans by up to 25% via activation of the DAF-16/FOXO-dependent pathway. The results indicate that chlorophyll is absorbed by the worms and is thus bioavailable, constituting an important prerequisite for antioxidant and longevity-promoting activities inside the body. Our study thereby supports the view that green vegetables may also be beneficial for humans.

Introduction

A diet containing fruits and green vegetables may counteract oxidative stress, ageing and health disorders, such as arteriosclerosis, diabetes, cancer, which are caused by high concentrations of reactive oxygen species (ROS) (Binstock, 2003; Finkel & Holbrook, 2000; Joseph et al., 1998).

Spinach (Spinacia oleracea) is among the widely consumed green vegetables in many countries of the world. It is thought to contain powerful natural antioxidants (NAO) which exhibit biological activity in in vitro and in vivo systems. Since many biological functions like protection from mutagenesis, carcinogenesis and ageing are due to antioxidant effects (Barja, 2002), spinach with a high level of antioxidants, such as flavonoids and p-coumaric acid derivatives (Bergman et al., 2001), has attracted considerable attention and is now subject to extensive investigations. Previous research showed that spinach has very high chemopreventive potential, enabling it to reduce the risk of developing several types of cancer (Boivin et al., 2009). In addition, long-term feeding studies using spinach extract were already carried out in rats, and the results indicated that spinach could be beneficial to counteract functional age-related deficits, and may even work against neurodegenerative diseases (Joseph et al., 1999; Kopsell et al., 2006).

The nematode Caenorhabditis elegans has been developed into an important model for biomedical research for several reasons: it is easy to culture, reproduces rapidly and prolifically with short generation times. Besides, it is a transparent sophisticated multicellular animal allowing the use of in vivo fluorescence markers. Many mutant strains of the nematodes are available containing GFP markers indicative of important cellular pathways, including those of oxidative stress tolerance, ageing and several diseases (Abbas & Wink, 2010; Henderson & Johnson, 2001; Hsu, Murphy & Kenyon, 2003; Link et al., 2015). Furthermore, C. elegans orthologues have been identified for 60–80% of human genes (Kaletta & Hengartner, 2006), thus making it a suitable model for human health conditions and diseases.

Although spinach has been promoted as a healthy vegetable (Chernomorsky, Segelman & Poretz, 1999; De Vogel et al., 2005), one of its obvious components, namely chlorophyll, has been neglected in most of available studies. We report here the antioxidative capacity of isolated chlorophyll from spinach, and discuss whether or not it can contribute to the potential benefits of spinach. We used C. elegans to investigate if natural chlorophyll mediates oxidative stress tolerance and promotes longevity. In addition, the in vitro antioxidant activity of chlorophyll was determined by DPPH and ABTS assays.

Materials and Methods

Isolation of chlorophyll from spinach

Spinach was purchased from local supermarkets (Spinat, Klasse II, L1907, Mählmann Gemüsebau GmbH, Cappeln, Germany) in 2014. The fresh leaves were ground in acetone using mortar and pestle. The green acetone extract was transferred into a separating funnel, which contained a 50% saturated NaCl solution in water and 50% n-hexane (Sigma-Aldrich GmbH, Steinheim, Germany). After gently shaking, the water phase was discarded. Saturated salt solution was used to wash the extract at least two times. Removal of the water via anhydrous sodium sulphate and filtration was performed before solvent evaporation using a rotary evaporator (Rotavapor R-300, BÜCHI Labortechnik GmbH, Essen, Germany).

Hexane was added to dissolve the dry pigment extract. Chlorophyll was isolated by means of liquid column chromatography using a column packed with silica gel as the adsorbent (silica gel 60, Merck, 0.063–0.200 mm; column: 3 × 25 cm) dispersed by hexane/acetone (9:1) solution (Sigma-Aldrich GmbH, Steinheim, Germany). The following eluents were applied: hexane/acetone 9:1, hexane/acetone 8:2, hexane/acetone 7:3, hexane/acetone 6:4, hexane/acetone 4:6, and acetone. Fractions were collected continuously. Each fraction was analysed by thin layer chromatography (TLC) using hexane/acetone 7:3 as mobile phase. Chlorophyll could easily be detected by its dark green colour. A UV spectrum (Biochrom WPA Biowave II) showed that the isolated chlorophyll was pure and did not contain other antioxidant secondary metabolites, such as flavonoids or carotene. Using column chromatography, we subsequently got 76.8 mg chlorophyll from 200 g spinach.

DPPH assay and ABTS assay

2,2-Diphenyl–1–picrylhydrazyl (DPPH∙) was used to measure the free radical scavenging activity of chlorophyll. (-)-Epigallocatechin gallate (EGCG) 95% (Sigma-Aldrich GmbH, Munich, Germany) was selected as a positive control. The method was described previously (Abbas & Wink, 2009) and performed with minor modifications. We prepared 0.2 mM DPPH∙ in ethanol, and added different concentrations of chlorophyll (0.1, 1, 5, 25, 100 µg/ml). The absorbance was measured at 517 nm after 30 min incubation in the dark. EGCG was applied in 5 concentrations (1, 2, 3, 4, 5 µg/ml). The ability to scavenge the DPPH∙ radicals was calculated by the formula: %Inhibition=A0−A1∕A0×100

where A0 is the absorbance of the control reaction (just DPPH∙ and ethanol) and A1 is the absorbance in the presence of substances.

The ABTS radical scavenging assay was conducted according to the method of our previous work (Sharopov, Wink & Setzer, 2015) with minor modifications. Briefly, 10 ml 7.0 mM ABTS+ free radical solution was added to 6.5 mg potassium persulfate and allowed to react for 16 h to form the stable ABTS+ radical cation. The solution was further diluted with absolute ethanol to obtain a final absorbance value about 0.7 ± 0.01 at 734 nm. Trolox (concentration range: 0.01–10 µg/ml) was applied as a positive control. The radical scavenging abilities of the chlorophyll (1–500 µg/ml) was determined by the reduction of the ABTS+ at 734 nm.

C. elegans strains and culture conditions

The wild type strain N2, and mutant strains TJ375 [hsp-16.2::GFP(gpIs1)], TJ356 [daf-16::daf-16-gfp;rol-6], GR1307 [daf16(mgDf50)] and CF1553 (muls84) were obtained from the Caenorhabditis Genetics Center (CGC, University of Minnesota, Minneapolis, MN, USA). All the strains were maintained at 20 °C on nematode growth medium (NGM) seeded with Escherichia coli OP50 as a food source as described previously (Abbas & Wink, 2009; Chen et al., 2013b).

Lifespan assay

Synchronized L4 larvae (three days after hatching) were transferred with a platinum wire to NGM agar plates (35 mm diameter) with about 60 worms per plate containing 0, 1, 10, 40, 80, and 100 µg/ml chlorophyll dissolved in absolute ethanol and diluted with liquid S-basal (Stiernagle, 2006) medium containing E. coli OP50 (Brenner, 1974). Equal amounts of solvent were used in all conditions (final concentration 0.5 % (v/v) ethanol). Living and dead worms were counted daily (starting at day 0 of adulthood) until all individuals were dead. Nematodes that failed to respond to a gentle touch were scored as dead. Nematodes suffering from internal hatch (hatching of embryos within the adult hermaphrodite) and those that escaped from the plates were censored. During the reproductive period, adult nematodes were transferred to new plates every day. After the reproductive period, adult worms were transferred to the new plates every 2 or 3 d. Wild type and strain GR 1307 [daf-16 (mfDf50)] were used for the life span assay.

Oxidative stress resistance

Isolated nematode eggs were transferred to Petri dishes containing liquid S-basal medium and E. coli OP50 as a food source. The worms were treated with 10 µg/ml chlorophyll the day after hatching, and an equal volume of solvent was added to the control group. All the worms were incubated at 20 °C for 3 days. Then, about 100 worms per group were transferred to the plates which contained 400 µM naphthoquinone juglone, a strong pro-oxidant which can induce lethal oxidative stress (Hassan & Fridovich, 1979). After 3 h exposure to acute oxidative damage, the survivors were scored.

In order to observe more details of the progress, the concentration of juglone was lowered, and the observation time was extended. The first day after hatching, the worms were treated for 72 h with various concentrations of chlorophyll (0, 1, 10, and 100 µg/ml). Approximately 120 worms per group were transferred into new plates with 200 µM juglone. Every one or two hours, dead worms were counted until all individuals from control group were dead.

Quantitation of Phsp-16.2::GFP and SOD-3::GFP expression

In the gpIs1 [Phsp-16.2::GFP] strain, Phsp-16.2 is expressed by either heat shock or oxidative stress. Age-synchronized L1 larvae were incubated in S-basal medium containing 10 µg/ml chlorophyll and E.coli OP50. After 48 h pre-treatment, the worms were exposed to 20 µM juglone as an oxidative stressor. After 24 h of treatment, worms were placed on microscopic slides in a drop of PBS containing a paralyzing agent (10 mM sodium azide). Strain CF1553 (muls84) was used to visualize SOD-3 expression. Similar methods as for TJ 375 were applied. L1 larvae of transgenic strain CF 1553 were treated with 10 µg/ml chlorophyll for 48 h. The expression of hsp-16.2 and SOD-3 was measured by quantifying the fluorescence of the reporter protein GFP. The intensity of fluorescence was analysed using ImageJ2X (ImageJ2X software; Rawak Software, Inc., Stuttgart, Germany). From each group 50 worms were randomly picked to measure the mean pixel density.

Subcellular DAF-16 localization

Depending on its activation/inactivation the transcription factor DAF-16 which plays an important role in the endocrine signalling pathways, is either localized in the cytosol or in the nucleus (Abbas & Wink, 2010; Baumeister, Schaffitzel & Hertweck, 2006; Chen, Rezaizadehnajafi & Wink, 2013). The TJ365 strain stably expresses a DAF-16::GFP fusion protein which can be detected by fluorescence microscopy in living worms. Age-synchronized L2 larvae were treated with 10 µg/ml chlorophyll for 1 h, while a heat shock positive control (37 °C for 30 min) and a non-treated negative control were carried out simultaneously. Subsequently, TJ356 worms were classified into three categories, (cytosolic, intermediate, and nuclear) after visual localisation of the DAF-16::GFP fusion protein by fluorescence microscopy (BioRevo BZ-9000, Keyence, Osaka, Japan).

Pharynx pumping rate

Pharynx helps worms for the ingestion of bacteria. Pumping rate is defined as the number of contractions of the pharyngeal terminal bulb in one minute. It can predict longevity and dietary restriction of C. elegans (Mango, 2009). The pharynx pumping assay were carried out followed the standard method (Wilkinson, Taylor & Dillin, 2012) in order to exclude the dietary restriction function of chlorophyll. To do so, wild type worms were transferred on the NGM agar plates contain OP50 under a light microscope. 10 worms per groups were randomly chosen to record the pumping rate by a hand-held counter from age day4 to day7.

Statistical analyses

Each assay was carried out in triplicate and statistical analysis was done in GraphPad Prism® software 5.01 (GraphPad Software, Inc., La Jolla, CA, USA) unless mentioned otherwise. Continuous variables were displayed as the mean ± SD. Statistical comparison of life span using a log-rank test was performed with Kaplan–Meier survival analysis by StatView 5.0 software (SAS Institute, Cary, NC, USA). Quantification of GFP expression of SOD-3 and HSP were assessed by a two-tailed unpaired student’s t-test between control and treated groups. Two-way analysis of variance (ANOVA) was performed to compare multiple groups in the the pharynx pumping assay.

Results

Free radical scavenging activity of chlorophyll

DPPH assay confirms that EGCG exhibits a strong radical scavenging activity, while the chlorophyll extract even at the highest concentration (100 µg/ml) was inactive in the concentration range measured (Fig. 1). According to the ABTS assay (Fig. 2), isolated chlorophyll exhibits free radical scavenging activity only in relatively high concentrations.

Figure 1 DPPH assay.

Free radical scavenging activity of the positive control EGCG (1–5 µg/ml) and chlorophyll (0.1 – 100 µg/ml) determined by the DPPH assay.

Figure 2 Reaction of trolox (0.01 – 10 µg/ml), chlorophyll (1–500 µg/ml) with ABTS+.

The reduction of the ABTS+ concentration is expressed as the absorbance at 734 nm.

Chlorophyll increases life span in wild type nematodes

N2 worms were exposed to different concentrations of isolated chlorophyll (0, 1, 10, 40, 80, and 100 µg/ml). Obviously, all the concentrations of the chlorophyll can significantly increase the longevity of C. elegans under normal conditions (Fig. 3 and Table 1). Up to a chlorophyll concentration of 40 µg/ml, the life span of C. elegans was prolonged in a concentration dependent manner. Higher concentrations also extended the mean life spans of N2, although the effect is less pronounced than those of lower chlorophyll concentrations which can increase the mean life span by 23.37 % and 25.96 % (p < 0.0001). However, no life span extension was observed in the daf-16 deficiency strain (Table 1 and Fig. 3D), which indicates that the anti-aging activity of chlorophyll might be related to the DAF-16 pathway.

Figure 3 Effect of different concentrations of chlorophyll C. elegans life span is presented by Kaplan–Meier curves.

The bold black lines present standard conditions, and the thin grey lines present chlorophyll conditions. (A) wide type, chlorophyll 1 µg/ml; (B) wide type, chlorophyll 10 µg/ml; (C) wide type chlorophyll 100 µg/ml; (D) daf-16strain, chlorophyll 10 µg/ml.

Table 1 Effect of chlorophyll on the life span of C. elegans.

Strain	Treatment	Concentration of chlorophyll (μg/ml)	N	Life span mean ± SE (d)	Life span extension %	p - value	
Wild type	Control	0	182	14.25 ± 0.22			
Plus chlorophyll	1	180	16.18 ± 0.08	13.54	<0.05	
	10	183	17.58 ± 0.09	23.37	<0.0001	
	40	60	17.95 ± 0.64	25.96	<0.0001	
	80	60	16.23 ± 0.64	13.9	<0.05	
	100	182	15.82 ±0.26	11.18	<0.05	
daf-16 (mgDf50)	Control	0	80	12.50 ± 0.20			
Plus chlorophyll	10	80	12.33 ± 0.19	−1.36	0.4727	
Notes.

N number of worms studied

Chlorophyll increases oxidative stress resistance

The L1 larvae were treated with a low concentration of chlorophyll (10 µg/ml) for 72 h before exposure to 400 µM juglone for 3 h. Juglone was applied as a pro–oxidant that is able to convert oxygen to the superoxide anion. Juglone thus generates intracellular oxidative stress and was used to investigate whether chlorophyll protects from juglone-mediated cell damage. The survival rate of nematodes pre-treated with chlorophyll increased by a maximum of 207.5% when compared to the control group (Fig. 4A). In order to gain a deeper understanding of the underlying mechanisms, the worms were fed with 3 different concentrations of chlorophyll (1, 10, and 100 µg/ml). The survivors were counted in intervals of 1 or 2 h after the exposure to 200 µM juglone (Fig. 4B). 100% of the worms in the control group deceased after 9 h under oxidative stress, while the chlorophyll-treated groups showed 80.17% (1 µg/ml), 87.18% (10 µg/ml), and 23.00% (100 µg/ml) viability. Therefore, we conclude that chlorophyll can significantly reduce the damage from juglone-induced acute oxidative stress, especially in low concentrations.

Figure 4 Effect of chlorophyll on reducing juglone-induced stress.

(A) Pretreatment of C. elegans with 10 µg/ml chlorophyll for 72 h protects the worms against acute oxidative stress induced by 400 µM juglone. (B) Survival of worms incubated with different concentrations of chlorophyll, and without pretreatment after application of 200 µM juglone.

Chlorophyll inhibits expression of Phsp-16.2::GFP

In order to further confirm that chlorophyll can protect the worms against oxidative stress, a chromosomally integrated transgenic strain gpIs1 containing the 400-bp hsp-16.2 promoter coupled to the gene coding for the green fluorescent protein (GFP) was employed in the experiment. After induction by juglone, a strong GFP expression in the pharynx of the worms could be detected, while no fluorescent signal was evident in the nematodes under normal culture conditions. We chose the best concentration (10 µg/ml) of chlorophyll from the oxidative stress resistance experiment for the further investigation. The GFP expression density in the pharynx of gpIs1 worms was reduced 40.53% after treatment with chlorophyll when compared to the non-treatment group. Figure 5 again demonstrates that chlorophyll mediates stress resistance in worms.

Figure 5 Induction of Phsp-16.2::GFP(gpIs1) reporter in response to juglone treatment.

gpIs1 worms were treated with 10 µg/ml chlorophyll for 48 h followed by 20 µM juglone induction for 24 h (B), while the control group was not subject to chlorophyll pretreatment (A). (C) Quantification of GFP expression. Data were from two independent experiments with 25 worms in each group. ∗∗∗p < 0.001.

Subcellular DAF-16 localization

The FOXO transcription factor DAF-16 has been identified to play a key role in the process of ageing, heat and oxidative stress resistance and other biological functions such as developmental arrest, fertility, fat storage, and metabolism (Henderson & Johnson, 2001; Hsu, Murphy & Kenyon, 2003). Since the nuclear localization of the transcription factor is an essential prerequisite for its bioactivity, we analysed the distribution of DAF-16 in the transgenic C. elegans strain TJ356 to detect whether chlorophyll is able to affect the location of DAF-16. Figure 7 shows that under normal culture conditions a cytoplasmic localization (67.86%) of DAF-16 is the predominant phenotype, while thermal stress (cytoplasmic localization in 16.67% of the worms) and chlorophyll (29.16% cytoplasmic localization) significantly increase the rate of nuclear and intermediate localization of DAF-16, which indicates that chlorophyll is capable to mediate the nuclear translocation of DAF-16.

Chlorophyll up-regulates SOD-3 expression

Sod-3 is a bona fide transcriptional target of DAF-16, including DAF-16 DNA-binding sites. Inhibition of insulin/IGF1 signalling can lead to the DAF-16-mediated up-regulation of sod-3, which can easily be visualized by a SOD-3::GFP reporter (Honda & Honda, 1999; Samuelson, Carr & Ruvkun, 2007; Zheng et al., 2013). Figure 6 shows that chlorophyll significantly increased the SOD-3 GFP expression density in strain CF1553, which was 32.66% higher than that of the untreated worms. SOD-3 is a downstream effector of DAF-16, which can be regarded as an oxidative stress-sensitive reporter to forecast the life span extension in C. elegans (Rea et al., 2005; Zheng et al., 2013). Hence, the anti-aging and oxidative resistance activities of chlorophyll depend on the DAF-16 dependent mechanism.

Figure 6 Chlorophyll induced up-regulation of SOD-3::GFP expression in strain CF1553.

(A) SOD-3 expression in the control group. (B) SOD-3 expression of chlorophyll pre-treated worms. (C) Quantification of GFP expression. Data came from two independent experiments with 25 worms in each group. ∗∗∗p < 0.001.

Figure 7 DAF-16 localization in TJ 356.

(A) Cytosolic localization; (B) intermediate localization; (C) nuclear localization; (D) effect of chlorophyll and thermal stress on the distribution of DAF-16 localization in TJ 356 shown in percentage. Three independent experiments with 20 worms per group were analysed.

Chlorophyll does not reduce the food intake of C. elegans

We analysed whether chlorophyll could exert its activities by dietary restriction (DR) due to reduced food intake. Degeneration and DR would result in a reduced motor activity in C. elegans; as a consequence pharyngeal pumping would be reduced (Huang, Xiong & Kornfeld, 2004; Samuelson, Carr & Ruvkun, 2007). Figure 8 clearly demonstrates that chlorophyll did not reduce the pharyngeal pumping rate, which is a measurement for detection of food intake of nematodes. Thus, the maintenance of pumping rates in nematodes treated with chlorophyll supports our assumption that chlorophyll exerts its activity not via dietary restriction.

Figure 8 Pumping rates of N2 from age day 4 to day 7.

No dietary restriction were observed. In contrast, chlorophyll assists to remain the pharyngeal function. ∗∗p < 0.01; ∗∗∗p < 0.001.

Discussion

Recent studies demonstrated that spinach extracts rich in phenols and flavonoids were able to prolong the lifespan and increase the resistance to both thermal and oxidative stresses in C. elegans via its free radical quenching function (Fan et al., 2011). The action resembles the effect of epigallocatechin gallate (EGCG) from green tea (Abbas & Wink, 2008) and anthocyanins from purple wheat (Chen et al., 2013a; Chen, Rezaizadehnajafi & Wink, 2013), which diminish oxidative stress and reactive oxygen species (ROS) levels in vivo by activation of the PI3 kinase/AKT pathway. According to our results from long-term and short-term lifespan experiments, chlorophyll, which has not been studied in this context before, also substantially reduces mortality of worms under either normal or oxidative stress conditions (Table 1, Figs. 3 and 4).

We know that chlorophyll shows red fluorescence under ultraviolet light. After being exposed to chlorophyll, the specific red fluorescence was observed in the intestine and anus of worms under the UV light; such a fluorescence was absent among untreated worms (Fig. S1). Therefore, we could confirm that chlorophyll was taken up by animals. Also, we applied broth microdilution assays and growth kinetics to determine that chlorophyll does neither impair nor enhance growth of E. coli OP50 (data not shown). Hence, the food source was not modified by chlorophyll. Furthermore, the pharyngeal pumping assay suggests that chlorophyll does not act via dietary restriction as the pumping rate would be reduced under DR conditions (Fig. 8). In addition, the size of the worms did not change under chlorophyll treatment (data not shown). It is reasonable to assume that chlorophyll directly causes longevity and resistance to oxidative stress of the worms.

Previous research indicated that chlorophylls can act as free radical scavengers by reduction of DPPH in a dose—dependent manner (Hsu et al., 2013). However, we could not reproduce these results using a chlorophyll extract of spinach which showed no activity in the DPPH system and low activity in the ABTS system in the applied concentration range (Figs. 1 and 2). It has been suggested that chlorophyll a exhibits considerable antioxidant activity only at high concentrations of about 1 mM (Lanfer-Marquez, Barros & Sinnecker, 2005) which were not used in our experiments. Whereas our ABTS assay results agree with several former reports, which suggested that chlorophyll provides protection by preventing autoxidation via a hydrogen donation mechanism breaking the radical chain reactions, the intact chemical structure of porphyrin seems to be essential for antioxidant activity (Endo, Usuki & Kaneda, 1985a; Endo, Usuki & Kaneda, 1985b).

In addition, other authors claim that synthetic metallo–chlorophyll derivatives, especially Cu- chelated compounds (Ferruzzi et al., 2002) have much higher antioxidant activities than the natural chlorophylls and Mg-free derivatives which showed almost no antioxidant capacity. These authors suggested that chlorophyll occurring in high levels in plants may play a role in chloroplasts protecting them against lipid oxidation (Lanfer-Marquez, Barros & Sinnecker, 2005). Some scientists even assumed that chlorophyll acts as a pro-oxidant when exposed to light, while the opposite might be true in the dark (Endo, Usuki & Kaneda, 1985a; Endo, Usuki & Kaneda, 1985b). As seen in our experiments, we could not detect a strong antioxidant effect in vitro, but a pronounced activity in vivo. This suggests that chlorophyll taken up by the worms, was metabolized into a more active antioxidant metabolite. This needs to be explored in future studies.

As discussed by other authors, antioxidant activity alone cannot explain longevity (Halliwell, 2012; Sadowska-Bartosz & Bartosz, 2014). For instance, resveratrol from red wine and other food items, known as an active free radical scavenger and hydrogen donator, can reduce acute oxidative damage in C. elegans, but is not capable of extending life span in C. elegans (Chen et al., 2013b). The similar finding has been reported for vitamin C and vitamin E as well (Ernst et al., 2013; Pallauf et al., 2013). Like resveratrol, chlorophyll apparently is able to protect C. elegans against oxidative stress. In contrast to resveratrol, chlorophyll is also able to significantly enhance the longevity of C. elegans under normal conditions. Thus, the life-span enhancing effect of chlorophyll is likely to not be based on the reduction of oxidative stress alone. However, it has been suggested that lifespan extension and stress resistance go hand in hand, and that they are probably regulated under a similar genetic program (Baumeister, Schaffitzel & Hertweck, 2006; Tatar, Bartke & Antebi, 2003).

Stress resistance and lifespan extension are evolutionarily conserved in many organisms, and mostly the DAF-16/FOXO-dependent pathway seems to be involved (Baumeister, Schaffitzel & Hertweck, 2006): by decreasing insulin/IGF-1 signalling (Gami & Wolkow, 2006; Tatar, Bartke & Antebi, 2003), or increasing JNK signalling (Oh et al., 2005) longevity and stress resistance can be enhanced. Treatment of worms with chlorophyll resulted in the translocation of DAF-16 from cytosol to nucleus, and its nuclear localization is a prerequisite for the transcriptional activation of signalling pathways. This activity might explain our results, but remains to be corroborated by more detailed experiments of the regulation of the genes involved. Furthermore, the up-regulation of the expression of SOD-3 and maintenance of pharyngeal function by chlorophyll are also in line with this point of view.

Conclusions

In conclusion, our experiments demonstrate that chlorophyll exhibits substantial antioxidant activity in vivo, and significantly improves antioxidant resistance of C. elegans. More important, chlorophyll can enhance the life span by over 20%. We assume that the protective capacity of chlorophyll might be attributed to its antioxidant activity and/or modulating activity of the DAF-16/FOXO-dependent pathways. Therefore, dietary chlorophyll derivatives not only benefits C. elegans, but also supports the recommendation of nutritionists to eat green vegetables and salads containing high contents of chlorophyll, as this could also help to improve human health and prevent diseases.

Supplemental Information

Figure S1 Supplemental Detection of chlorophyll inside worms

Click here for additional data file.

Data S1 Raw data for DPPH assay

Click here for additional data file.

Data S2 Raw data of pumping assay

Click here for additional data file.

Data S3 raw data for ABTS assay

Click here for additional data file.

Data S4 Raw data of SOD-3 expression

Click here for additional data file.

Data S5 Raw data for quantitation GFP

Click here for additional data file.

Markus S. Braun kindly helped to improve the manuscript.

Additional Information and Declarations

Competing Interests

Author Contributions

Data Availability

Michael Wink is an Academic Editor for PeerJ.

Erjia Wang conceived and designed the experiments, performed the experiments, analyzed the data, wrote the paper, prepared figures and/or tables.

Michael Wink conceived and designed the experiments, contributed reagents/materials/analysis tools, wrote the paper, reviewed drafts of the paper.

The following information was supplied regarding data availability:

Raw data has been added as supplemental data.

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
