# Peer review of "Chlorophyll enhances oxidative stress tolerance in Caenorhabditis elegans and extends its lifespan"

_PeerJ, doi:10.7717/peerj.1879_

## Round 0.1 · original submission · Major Revisions

Some concerns have been raised about whether your statistical methods are all appropriate and whether they are described insufficiently. Please consult with a professional statistician and have him or her carefully review this and help as needed.

Please also be sure your reporting conforms to the ARRIVE guidelines (https://www.nc3rs.org.uk/arrive-guidelines).

Reviewer 1 ·

Basic reporting

no comments

Experimental design

authors should list statistical tests performed. There is an issue with experimental design related to the anesthetic apparently used for two assays (see detailed comments).

Validity of the findings

A test of direct activity of the chemical on the organism should be performed. This can be done by examining the chemicals effects on worms co-cultured with killed or UV treated bacteria. This will supplement (not replace) the authors results with live bacteria.

Additional comments

Wang and Wink explore the pharmacological effects of chlorophyll on C. elegans stress resistance and lifespan. The authors demonstrate that treatment with chlorophyll extends the lifespan of worms and promotes resistance to oxidative stress. The authors test the antioxidant activity of chlorophyll in vitro and find that it has weak activity at the concentrations found to be effective in vivo. They cite references and speculate in the discussion that the in vivo form may have more significant antioxidant activity. The experiments in this paper are generally well controlled and executed. The authors also generally stick to reasonable interpretations of their data, are concise in their descriptions and the paper reads quite well.
Major criticisms:
1. The authors have not convinced me of a relationship between the effects they see with chlorophyll and DAF-16 for several reasons:
First, the anesthetic sodium azide should not be used with TJ356 as sodium azide induces the nuclear localization of the DAF-16::GFP (in that strain at least), and stress responses possibly including induction of HSP-16.2. It is likely this occurs by sodum azide disrupting cytochrome oxidase which may result in the generation of oxidative stress, the very phenomenon being studied. This makes sodium azide a poor choice of anesthetic. The authors do not directly mention what anesthetic is used in the DAF-16 nuclear localization experiments. However, from the images it would appear they use one. Since the only one described in the methods is sodium azide, I am forced to conclude that this was used in these experiments as well as in the gpIs1 experiments (for which it is also likely an inappropriate anesthetic). These experiments should therefore be re-done with a different anesthetic (such as levamisole) or be removed from the manuscript. Second, the authors imply that the mechanism of chlorophyll’s effects on lifespan and stress resistance is through DAF-16. They could directly test this by repeating their stress resistance and lifespan experiments with a null allele of daf-16, such as the deletions mu86 or mgDf47. Additionally, they could test for activation of DAF-16 with QPCR on a transcriptional target such as SOD-3 or even use imaging such as they use in the manuscript on the Psod-3::GFP reporter strain. These would complement or replace the DAF-16::GFP r3esult in the current manuscript.
2. The authors should test for the effects of chlorophyll using killed or UV treated bacteria. I am of the opinion that any new manuscript describing lifespan extension in C. elegans from a chemical should define whether the effect is dependent on the presence of live co-cultured bacteria. That is an important issue and should not be left for other researchers to determine.
3. Does chlorophyll get into the worms? They could exhibit characteristic fluorescence, if it is. Perhaps you could approximate its in vivo concentration based on the fluorescent signal. Or at least demonstrate that it gets into the animal and is not just passed through the lumen of the intestine.

Minor points and notes:
The statistical tests used to determine the p values need to be described for the results in figure 4, 5 and 6.
In all relevant figure legends the type of error bars used should be defined.
The authors use the term, “antioxidant stress resistance” several times in the paper including the abstract. Perhaps this is an oversight, but it could be taken to mean that the chemical compound is somehow acting to improve antioxidant activity, or is itself acting as an antioxidant, which is not demonstrated in this manuscript. This should therefore be changed to “resistance to oxidative stress” or some other phrase that defines the outcome and not the mechanism.
The authors mainly use the spelling ageing but in line 28 of pg3 they use the spelling aging.
The lifespan assay methods section needs improvement. The first sentence is not clear and should be improved and broken into at least two sentences; “L4 of the following generation”, I am not sure what this means. Following who? Generation G1? Convention is P0 (for parental) and all other generations are referred to as f1, f2 f3…etc. Please correct the section to answer how the population being assayed was synchronized for the lifespan assay (With an egg lay or hypo chlorite treatment, or were they picked at a specific stage?).
Line 129 should be “a variety of” or “various”, instead of “variety”.
The methods section describing the use of TJ375 should be improved for clarity and to adhere to standard nomenclature practices. There are multiple inconsistencies and typos in this section including in the title, which should be different from the title in the results section (which has a typo in it). The transgene in the strain TJ375 can be referred to as gpIs1[Phsp-16.2::GFP]. You should not use the forward slash which can imply something else. It is important to denote that the transgene drives GFP but not the HSP-16.2 protein, I prefer the capitol P in front but other methods are acceptable. It should be clear that you are quantifying a GFP surrogate and not HSP-16.2 throughout the manuscript, while it is clear in the figure legend, it is not in the methods section or in the title for the methods and results sections.
Line 148 be specific; not “as mentioned before.”
Line 149, this is the first mention of “bleaching” and should be explained.

·

Basic reporting

No comments

Experimental design

See general comments. Further evidence whether the effects of chlorophyll are direct is needed to support conclusions. Additional controls and experiments can improve this.

Validity of the findings

See general comments. The conclusions might be correct, but should be tested more rigorously to rule out some obvious alternative hypotheses. Additional data would strengthen or modify conclusions, either of which would be sufficient.

Additional comments

The manuscript, titled “Chlorophyll enhances oxidative stress tolerance in Caenorhabditis elegans and extends its lifespan,” describes results obtained largely from dosing the nematode C. elegans with the common plant compound, chlorophyll. The authors show that addition of chlorophyll to worm plates increases lifespan and improves resistance to at least one oxidative stressor (juglone), and suggest this means that chlorophyll has a role as an antioxidant and anti-aging compound. The authors also show that chlorophyll does not scavenge DPPH, as previously reported, but does scavenge another pro-oxidant, ABTS, and causes nuclear relocation of the FOXO transcription factor, DAF-16, in worms.

The results from the paper are intriguing, but in order to support their conclusions, there several basic experiments that should be performed to validate the results. These experiments and comments are listed below. My major concern is whether chlorophyll is directly affecting the worms at all. The increase in stress resistance, longevity and daf-16 nuclear localization all resemble certain types of dietary restriction (e.g. sDR, see EL Greer et al. 2009, Aging Cell). It is easy to hypothesize that chlorophyll simply causes the worms to eat less. They also resemble, to some extent, the worm response to certain osmotic stresses (e.g. sorbitol, where the dose respose looks similar), and both of these types of responses are not considered by the data. In order to claim that chlorophyll does, in fact, directly cause increases in stress resistance and longevity, further supporting experiments are required

Major Issues/questions:
1) Does chlorophyll get taken up by the worms? The results do not show whether the worms actually ingest the chlorophyll and thus whether the actions are more likely to be through chlorophyll bioactivity. This should be easy to test and likely has already been done. The abstract directly states "The results indicate that
chlorophyll is absorbed by the worms and thus is bioavailable, constituting an important prerequisite for antioxidant and longevity promoting activities inside the body." I do not see the data that support this statement.

2) Do the bacteria take up the chlorophyll? One possible response to chlorophyll on the plate is to modify the bacterial lawn on the plate, thereby modifying the food source of the worms. Also easy, and can be done quickly.

3) As described above, these results look similar to other reports on worms exposed to dietary restriction (i.e. increased longevity, stress resistance, and DAF-16 activity). Without knowing whether chlorophyll is actually taken up and active in the worm/makes the bacteria less tasty/nutritious, it is hard to know whether the results support the claims. Even if chlorophyll is taken up by the worms, its main role could still be to slow digestion or curb appetite (perhaps worms don’t like vegetables either!). One easy way to test this is to verify whether chlorophyll is additive with DR.

4) The proper controls for the chlorophyll plates should realistically also include another compound, similar in size (preferably known to be inert), at the same concentration as the chlorophyll to rule out osmotic effects. Vehicle only leads to different osmolarity among the plates.

5) The statement that chlorophyll acts as an antioxidant while also activating daf-16 seems incongruent. Oxidative stress is known to activate daf-16, and daf-16 activation would explain resistance to juglone and increased longevity. If chlorophyll activates daf-16, it is likely through increasing oxidative stress or from some other stress (like DR). Also, validating whether daf-16 is required for the longevity/stress resistance phenotype would strengthen the conclusions.

Minor suggestions include combining figures 1 and 2 and lengthening/fleshing out the results section further, but are relatively trivial in the bigger picture.

·

Basic reporting

The article is written in good clear English. The figures are clear and self-explanatory. The Introduction and The Discussion serve their purpose to place article’s finding in a broader scientific context. The context chosen may, though, be somewhat misleading (see below). While review articles may generally be appropriate as references, 10 to 15-year-old review articles cited in the Introduction are likely to be obsolete and appropriate references should be updated.

Experimental design

Methodologically, the article looks mostly sound. The choice of spinach as a source of chlorophyll is somewhat questionable though. Chlorophyll is present in all green plants, but among them, the authors chose to extract it from spinach. According to the authors’ citations, spinach extract has a high antioxidant activity and this activity is attributed to some well characterized substances and, possibly, to some unknown substances. Thus, dealing with chlorophyll purified from such a rich in antioxidants plant as spinach will always keep open an issue of a low abundance, but high activity contaminant that “explains” chlorophyll antioxidant properties.

Validity of the findings

While the finding described in the article seem to be valid and statistically supported, their interpretation may require some re-interpretation.
The authors showed in two in vitro assays that chlorophyll does not seem to have strong antioxidant properties. One explanation proposed by the authors for an apparent contradiction between the lack of an in vitro antioxidant activity and an apparent in vivo antioxidant protection was that chlorophyll might have metabolized in vivo into an active antioxidant. However, the other mechanism for chlorophyll antioxidant activity, through activating the daf-16 pathway, seems to be better supported by the data. Activation of the Daf-16 FOXO pathway explains not only increased resistance to oxidative stress, but also extended life span. However, the daf-16 pathway is usually activated by environmental stresses (e.g. overcrowdedness, high temperature, oxidative stress, etc). The phenomenon of hormesis, i.e. stimulating stress resistance by a pre-treatment with a low dose of a stressor, has been well documented for C. elegans and the entire story described in the article seems to fit into the hormesis framework. If this is the case, chlorophyll works not as an antioxidant, but rather as a mild “poison”, triggering C, elegans protective pathways. The mechanism of this action is still to be explored and its relevance to higher animals/humans is to be determined.

---

## Round 0.2 · Minor Revisions

Decision: Accept pending minor revisions.

Reviewer 1 ·

Basic reporting

Please include numbers of worms and replicates tested from lifespan analysis.

Experimental design

While the whole of the article certainly warrants publication. I am unsatisfied with the experiments performed to address my previous criticism of the authors’ failure to test whether live bacteria are required for the observed effects of chlorophyll on C. elegans. That question remains unanswered. The experiments performed in the re-submission are insufficient to specifically address whether the live bacteria are the target of chlorophyll effect. In that case, the chlorophyll could alter the bacteria in some manner (not necessarily altering growth rate) which then has the secondary effect of promoting longevity and stress resistance in the worms, due solely to the action of the chemical on the bacteria (this possibility is not noted in the authors' rebuttal). This phenomenon has been previously observed for the drug metformin. The experiment is not particularly difficult or time consuming, and would improve the manuscript.

Validity of the findings

It does not seem appropriate to conclude that worms are not induced into DR simply because their pumping rate is approximately normal. It would seem more prudent to conclude that the feeding rate is not grossly altered by the treatment.

·

Basic reporting

There are a few grammatically questionable sentences, but generally the article is easy to understand.

Experimental design

See comments

Validity of the findings

See comments

Additional comments

I am satisfied that the authors have proven that 1) chlorophyll gets taken up by worms, 2) bacteria do not metabolize large amounts of chlorophyll, and 3) chlorophyll does not change pumping rate. Generally, I am convinced of the findings, with a couple of caveats. First, the statement that maintenance of pumping rules out a DR affect from chlorophyll is too strong. To better show that chlorophyll and DR do not interact, the best experiment is to combine the interventions and observe whether they are additive. There are multiple ways to cause DR, and chlorophyll could plausibly change digestion or another non-pumping related aspect of feeding. If the authors want to make that statement, they should do this experiment. Alternatively, the authors could soften the statement to say something like "the maintenance of pumping rates when treated with chlorophyll is consistent with chlorophyll's effects being distinct from dietary restriction." The other area of concern is the stress resistance assay. It is unclear whether chlorophyll is increasing resistance to ROS/juglone by activating daf-16 (probably), scavenging ROS (maybe), or both. This could easily and quickly be answered by testing whether chlorophyll benefits the daf-16 mutant strain in juglone, and likely has already been done in some form. I do not think this assay is crucial to support the basic finding of the paper, but would help clarify the seemingly two major hypotheses presented, that Chlorophyll acts through daf-16 and that chlorophyll is, by itself, an antioxidant.

·

Basic reporting

Satisfied

Experimental design

Satisfied

Validity of the findings

Satisfied

---

## Round 0.3 · accepted · Accept

Thank you for addressing the minor revisions requested in the prior version.